**Data Availability Statement:** The data set necessary to validate our analysis contains identifying patient-level data which cannot be suitably de-identified or aggregated and can

# An exploration of under-registration of chronic kidney disease stages 3–5 in Belgian general practices using logistic regression

**Ine Van den Wyngaert**[ID]*, **Pavlos Mamouris**[☯], **Bert Vaes**[☯], **Gijs Van Pottelbergh**[☯]

Academic Centre for General Practice, Department of Public Health and Primary Care, University of Leuven, Leuven, Belgium

☯ These authors contributed equally to this work.

* ine.vandenwyngaert@kuleuven.be

## Abstract

### Background

Early detection and treatment of chronic kidney disease (CKD) can prevent further deterioration and complications. Previous studies suggested that the diagnosis is often made when advanced renal failure occurs. The aims of this study were to describe the prevalence of unregistered CKD stages 3–5 in a Belgian General Practitioner population, to determine risk factors for under-registration and to investigate the diagnostic delay.

### Methods

The analyses were carried out in the INTEGO database, a Flanders general practice-based morbidity registration network. The study used INTEGO data from the year 2018 for all patients ≥18 years old. CKD was defined as two consecutive eGFR laboratory measurements (eGFR <60 mL/min/1.73m$^2$) at least three months apart during the baseline period. Registered CKD was characterised by a documented diagnosis of CKD (ICPC2 U99) during the ≥12-month lookback period before the first eGFR measurement and up to six months after the second eGFR in the EHR. The prevalence of unregistered CKD and the median time of diagnostic delay were estimated. Baseline characteristics were described. A multivariate cross-sectional logistic regression analysis was conducted to identify determinants of unregistered CKD. We estimated the odds ratios and their 95% confidence interval.

### Results

Among included patients, there were 10 551 patients (5.5%) meeting the criteria of CKD. The prevalence of unregistered CKD was 68%. The mean diagnostic delay was 1.94 years (Standard deviation 0.93). Being a male, a concurrent diagnosis of diabetes, stroke, heart failure and hypertension, and more severe CKD (stages 3b, 4 and 5) independently increased the chance on registered CKD.

therefore only be accessed inside a monitored analysis environment. These restrictions were imposed by the Belgian National Information Security Committee: section Social Security and Health. Websites (only FR and NL): -https://ehealth.fgov.be/ehealthplatform/nl/informatieveiligheidscomite -https://ehealth.fgov.be/ehealthplatform/fr/comite-de-securite-de-linformation Main deliberations (only FR and NL): - Beraadslaging nr. 15/009 van 17 februari 2015 betreffende de generieke methode voor de uitwisseling van gecodeerde persoonsgegevens die de gezondheid betreffen in het kader van healthdata.be en healthstat.be è -https://ehealth.fgov.be/ehealthplatform/file/view/23acd078e8a4cb025e216b6d24906ae480b57cf8?filename=15-009-n023-healthdata_be.pdf - Délibération n° 15/009 du 17 février 2015 relative à la méthode générique d'échange de données à caractère personnel codées relatives à la santé, dans le cadre de healthdata.be et healthstat.be. è https://ehealth.fgov.be/ehealthplatform/file/view/ea116a489cc473f8174c8559dd060b4faa3ff651?filename=15-009-f023-healthdata_be.pdf ° INTEGO deliberations (only FR and NL): - Beraadslaging nr. 13/026 van 19 maart 2013, laatst gewijzigd op 7 april 2020, met betrekking tot de mededeling van gepseudonimiseerde persoonsgegevens die de gezondheid betreffen, via het healthdata.be platform, in het kader van de samenstelling, het gebruik en de ter beschikking stelling van een register voor epidemiologisch onderzoek (Intego). è https://ehealth.fgov.be/ehealthplatform/file/view/AXGB3PooVF2eEzQeBL5P?filename=13-026-n162-INTEGO%20Covid19gewijzigd%20op%207%20april%202020.pdf - Délibération n° 13/026 du 19 mars 2013, modifiée en dernier lieu le 7 avril 2020, relative à la communication de données à caractère personnel pseudonymisées relatives à la santé, via la plateforme healthdata.be, dans le cadre de la constitution, de l'utilisation et de la mise à la disposition d'un registre de recherches en épidémiologie (Intego). è https://ehealth.fgov.be/ehealthplatform/file/view/AXGB2dtUmTlaOSp4Nmk3?filename=13-026-f162-INTEGO%20Covid19-modifi%C3%A9e%20le%207%20avril%202020.pdf Data requests may be sent to Mr. Roel Heijlen, Data Protection Officer healthdata.be (Sciensano), Roel.Heijlen@sciensano.be, Rue Juliette Wytsmanstraat 14, 1050 Brussels. Interested researchers will need to provide their name, first name, professional organization name, email address, mobile number and the database of interest to request access.

**Funding:** - This study was funded by AstraZeneca BeLux (https://www.astrazeneca.be/). - AstraZeneca discussed the study design, but did

## Conclusion

The proportion of patients who had no registered CKD code in the EHR was substantial. The differences between registered and unregistered patients make thinking about solutions to facilitate registration in the EHR imperative.

## Introduction

The ageing and growth of the world's population have led to an increasing number of people living with chronic diseases [1]. Chronic kidney disease (CKD) is defined as structural or functional kidney damage that is subdivided in different stages according to their estimated glomerular filtration rate (eGFR) values and albuminuria category [2]. The global prevalence of CKD is consistently estimated to be between 11 and 13%, with the majority of CKD patients in stage 3 (eGFR 30–59 ml/min/1.73 m$^2$) [3]. CKD reflects a complication of many different diseases, including hypertension and diabetes [4–6]. It is also associated with an increased risk for cardiovascular disease and premature death [4, 7, 8]. High health care costs are attributed to CKD. For the period 2017–2019, the National Institute for Health and Disability Insurance (NIHDI) reported an average annual expenditure of €434 464 745 on dialysis, with an average of 13 562 interventions per year performed [9]. For the same period, an average of 389 kidney transplants per year were documented with an average expenditure of €1 017 640 per year [10].

People with CKD experience few symptoms in the early stages, but are at risk for impaired quality of life in end-stage kidney disease. Unfortunately, some studies suggest that, despite a growing worldwide attention on the burden of CKD, a significant proportion of patients with early-stage CKD remains without official diagnosis and the diagnosis of CKD predominantly occurs during advanced disease (stages 4 and 5) [11–14]. At these stages there are limited opportunities to prevent further deterioration and to avoid complications [4, 8, 15]. Early detection and treatment of CKD is necessary, because it can influence further deterioration of kidney function and it reduces the risk of complications and the number of patients developing End Stage Renal Disease (ESRD) [16, 17]. Physicians in primary care have a key role in the early detection and management of CKD through lifestyle advice and drug adjustments [18]. If active screening and monitoring in risk populations is conducted consistently, early disease identification can also lead to significant cost savings [16].

It is unclear whether there are observable differences between registered and unregistered patient groups. Moreover, the demographic and clinical predictors of unregistered CKD remain to be determined. If the diagnosis is not registered in the Electronic Health Record (EHR), then it probably will not be mentioned in referral letters to specialists, which may be important in providing continuity of care. This study aimed to describe the prevalence of unregistered CKD stages 3–5 in a Belgian General Practitioner (GP) population. We examined whether certain patient groups were more likely to stay unregistered. Furthermore, we investigated the diagnostic delay (time between abnormal eGFR and diagnosis).

## Methods

### Study setting and data source

The analyses were carried out in the INTEGO database, a Flanders general practice-based morbidity registration network at the Department of General Practice of the University of

not play a role in the data collection, data analysis, decision to publish or preparation of the manuscript. - The grant was given to the research group, not to an individual author.

**Competing interests:** The authors have declared that no competing interests exist.

Leuven. INTEGO is a large database as the result of continuous recording in a network of general practices since 1994. This is the only operational computerised morbidity registration network in Belgium based on more than 100 general practices data. Only 86 practices, represented by 454 GPs, who provided the best data according to our algorithm were used.

Data collection is regulated by an opting-out procedure. INTEGO procedures were approved by the ethical review board of the Medical School of KU Leuven (N° ML 1723) and by the Belgian Privacy Commission (no SCSZG/13/079). Written consent was obtained from every of the participating Intego primary care practices. GPs applied for inclusion in this registry. Before acceptance of their data, registration performance was audited using algorithms to compare their results with those of all other applicants. Only the data of the practices with optimal registration performance were included in the database. The INTEGO GPs prospectively and routinely registered all new diagnoses and new drug prescriptions, as well as laboratory test results and patient information, using computer-generated keywords internally linked to codes.

Patient characteristics and diagnoses are encoded in the INTEGO registry. Diagnoses are coded using a very detailed thesaurus and classified using the International Classification of Primary Care (ICPC-2; WHO FIC Collaborating Centre). Furthermore, all laboratory tests performed by GPs are included in the database. Drugs were classified according to the WHO's Anatomical Therapeutic Chemical (ATC) classification system (WHO FIC Collaborating Centre).

The methodology of data collection, study design, and analyses have been previously reported [19].

## Study population

International and local guidelines for CKD management recommend that patients should be diagnosed with CKD if the reduction in kidney function [estimated glomerular filtration rate (eGFR) $<$60 mL/ min/1.73 m$^2$] is present for more than three months [2, 17, 20]. All patients $\geq$18 years old with two consecutive eGFR laboratory measurements indicating CKD (eGFR $<$60 mL/min/1.73m$^2$) recorded at least $>$90 and $\leq$730 days apart during the baseline period were included in the study and constituted the denominator for the calculation of the prevalence of unregistered CKD. EGFR was calculated using both MDRD and 2009 CKD-EPI equation, as is preferred in the KDIGO guideline [2]. The date of the second qualifying eGFR was considered as the index date. The current study used INTEGO data from the year 2018. The end date of follow-up of the data was 31/03/2022. Selected patients had at least one eGFR measurement $<$60 mL/min/1.73m$^2$ in 2018 and belonged to the GP's yearly contact group.

**Unregistered CKD case definition.**   Patients with unregistered CKD were identified if they had no diagnostic code for CKD (ICPC-2 U99) at any time during the $\geq$12-month lookback period before the first eGFR measurement and up to 6 months post-index date (Fig 1). Those with a documented diagnosis of CKD (U99) during this time period were considered as having registered CKD. We manually checked both the code and the written diagnosis whether the code did merge with CKD. It was assumed that patients with at least one diagnostic code for CKD during the above specified time window had registered CKD.

## Study selection criteria

**Inclusion criteria.**   * CKD population: a description can be consulted in the study population section.* At least 12 months of continuous presence in the database or registration in the data prior to the first qualifying eGFR.

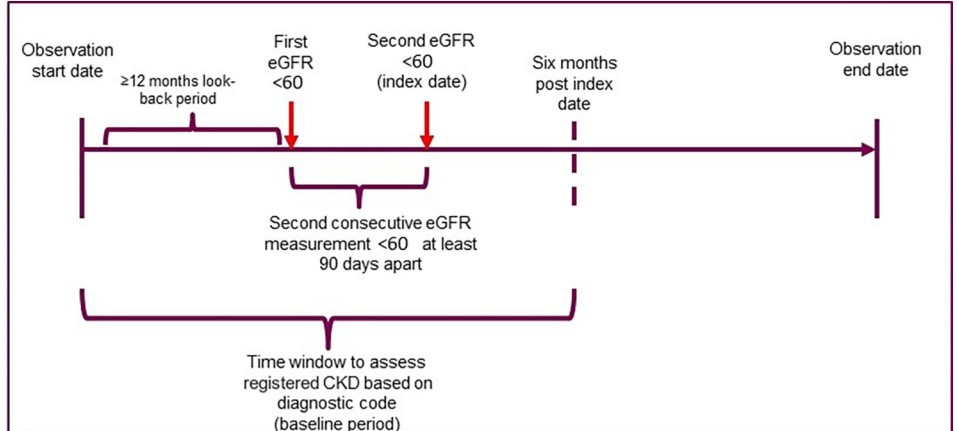

**Fig 1. Study timeline.**

**Exclusion criteria.** Patients were excluded if they had a solid organ transplant (ICD10 Z94.9) before the index date.

## Statistical analysis

All analyses were performed using R software, version 4.0.4. The first part of the study was descriptive in nature. The prevalence numerator included all patients who fulfilled the unregistered CKD case definition at any time during the baseline period. The prevalence denominator included all individuals meeting the inclusion criteria for this study (with and without documented diagnosis of CKD [U99]). Prevalence was estimated by dividing the numerator by the denominator. To assess the prevalence of unregistered CKD, patients were followed from the index date until six months post index date or observation end date (due to death, transfer out of the database or end of date coverage), whichever came first. The diagnostic delay was assessed in unregistered CKD patients who received a CKD diagnosis at later stage. The diagnostic delay was calculated as the difference of the date of U99 code received in the ≥6-month post-index date period minus the index date. Then, the median time was calculated. Follow-up was stopped at the time of CKD diagnosis or observation end date (due to death, transfer out of database or end of data coverage period).

Baseline characteristics (prior to the index date) were described for the study population. Medical history and comorbidity were assessed using all available data prior to the index date. For baseline assessment of physiological and laboratory values, the most recent assessment prior to the index date during the 12-month pre-index period was used. Similarly, medication use was ascertained within 12 months before the index date. Medication prescription was considered as used by the patient if at least one prescription could be found. Categorical variables were summarised using patient counts with percentages, and continuous variables were summarised using means with standard deviations (SDs). The chi-square was used for categorical variables and two-sample t-test for continuous ones. P values less than 0.05 were considered significant.

Besides, a multivariate cross-sectional logistic regression analysis was conducted to evaluate the associations of different variables with registered CKD [21]. We estimated the odds ratios (OR) and their 95% confidence interval (CI). P values less than 0.05 were considered significant. Only smoking was a missing variable. We imputed the smoking variable for the year 2018, using the method of Multiple Imputation as described by Rubin and implemented in

Rpackage mice by Van Buuren [22, 23] using several auxiliary variables including age, gender, socioeconomic status, and relevant diagnoses and prescriptions.

# Results

## Prevalence and diagnostic delay

After extraction from the INTEGO database, 231 702 patients ≥18 years old were detected in 2018 (Fig 2). 40 216 patients were excluded because the general practice didn't meet the criteria for best quality register. Among included patients, there were 10 551 patients (5.5%) with two consecutive eGFR laboratory measurements indicating chronic kidney disease (CKD) (eGFR <60 mL/min/1.73m$^2$), recorded at least three months apart during the baseline period. Out of them, 7176 patients had no diagnostic code for CKD (U99) at any time, during the ≥12-month lookback period before the first eGFR measurement and up to 6 months post-index date. Thereby, 5961 patients out of this group (83.1%) never received a diagnostic code, whereas 1215 patients (16.9%) were delayed in receiving a diagnostic code with a mean of 1.94 years (SD 0.93). The maximum follow-up period was 3.97 years. The mean diagnostic delay was the longest in CKD stage 4 (2.10 years) and it was 8.4 months longer than in CKD stage 5 (1.40 years) (Table 1).There were 106469 patients (55.6%) without any eGFR measurement and 1691 (0.9%) with only one. In this group, no estimate of renal insufficiency could be made. The prevalence of unregistered CKD was 68%, of which 17% received a delayed diagnosis (Fig 2). Tables 1 and 2 respectively describe the prevalence of CKD according to the CKD stage and the age group. The majority of both registered and unregistered CKD patients was in stage 3 and was over 75 years old. In stage 5 and age group >75 were the largest proportions of registered (75.7% in stage 5 and 34.4% in age group >75) compared to unregistered within their groups. Fig 3 shows the spread of the diagnostic delay.

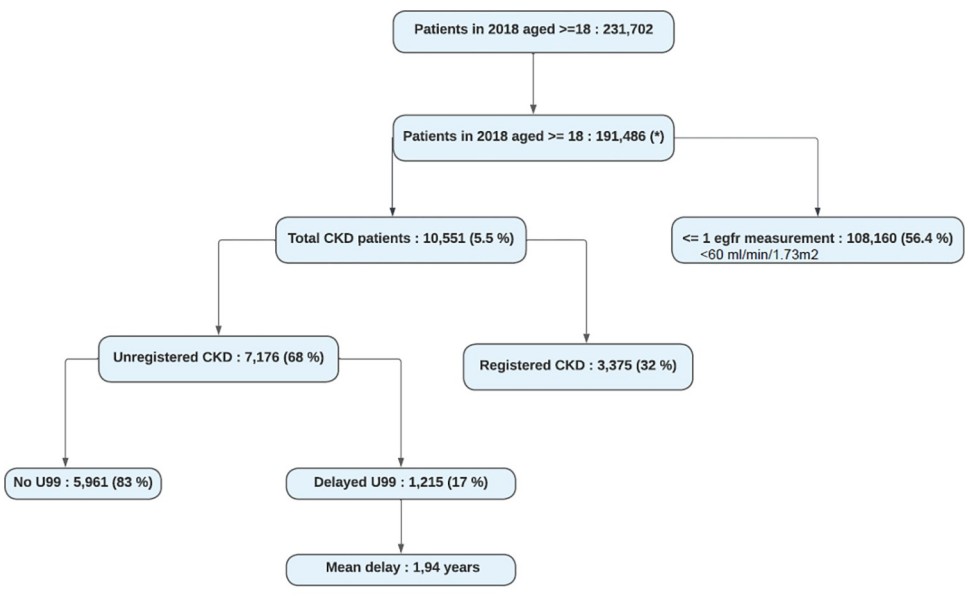

(*) Inclusion of the best quality register GPs

**Fig 2. Prevalence and diagnostic delay.**

**Table 1. The prevalence of CKD and time to diagnosis, according to the CKD stage.**

| Variable | Registered CKD | | | Unregistered CKD | | | Unregistered CKD patients with delayed diagnosis | | | Mean delay (years) |
|---|---|---|---|---|---|---|---|---|---|---|
| | A | B | C | A | B | C | A | B | C | |
| Total | 3375 | 100% | | 7176 | 100% | | 1215 | 100% | | |
| Stage 3a | 1762 | 52.2% | 22.9% | 5945 | 82.8% | 77.1% | 886 | 72.9% | 11.5% | 1.94 |
| Stage 3b | 1181 | 35.0% | 52.8% | 1056 | 14.7% | 47.2% | 299 | 24.6% | 13.4% | 1.92 |
| Stage 4 | 354 | 10.5% | 70.2% | 150 | 2.1% | 29.8% | 29 | 2.4% | 5.8% | 2.10 |
| Stage 5 | 78 | 2.3% | 75.7% | 25 | 0.3% | 24.3% | 1 | 0.1% | 1.0% | 1.40 |

A) Number of patients per CKD stage.

B) Number of patients per stage in relation to the total number of registered, unregistered of delayed CKD patients, expressed in percentage (%).

C) Number of registered, unregistered and delayed patients per stage in relation to the total number of CKD patients per stage, expressed in percentage (%). Stage 3a (eGFR 45–59 ml/min/1.73 m$^2$), Stage 3b (eGFR 30–44 ml/min/1.73 m2), Stage 4 (eGFR 15–29 ml/min/1.73 m$^2$), Stage 5 (eGFR <15 ml/min/1.73 m$^2$).

## Patient characteristics at baseline

Patient factors, comorbidities and treatment were described at baseline (period prior to the index date) (Table 3). The mean age for both registered and unregistered was above 75 years. Average BMI measurement showed overweight in both registered and unregistered CKD groups. Regarding the laboratory results, mean eGFR was lower and mean uACR ratio was higher in the registered group compared to the unregistered at baseline. Proteinuria is considered as clinically significant from a value of >450 mg/g, which was not the case for the mean uPCR in both the registered and unregistered. In the registered group, more patients were in stage 3B, 4 and 5 than in the unregistered group. Hemoglobin, hematocrit, potassium, uric acid and cholesterol values were similar in registered and unregistered CKD groups. Only a small percentage of patients had severe potassium or hemoglobin abnormalities, although these results were not significant. Further, a slightly larger number of patients with registered CKD (52.6%) received an ACEi or ARB compared to the unregistered CKD patients (46.3%). Almost half of patients with a CKD code received Beta-blockers or Lipid lowering drugs. Differences in percentages were seen between the registered and unregistered groups for patients taking diuretics, antidiabetic drugs, lipid lowering drugs, platelet aggregation inhibitors and anticoagulants. Hypertension (ICPC2 K86) was more frequently present in the registered (64.4%) compared to the unregistered (51.7%). Further, 33.1% of registered patients had a diagnostic code of type 2 diabetes (T90) at baseline, whereas 28.2% of unregistered patients had. Since the U99 code is a collective code for multiple urinary tract diseases like chronic

**Table 2. The prevalence of CKD according to the age groups.**

| Variable | Registered CKD | | | Unregistered CKD | | | Total | |
|---|---|---|---|---|---|---|---|---|
| | A | B | C | A | B | C | A | B |
| Total | 3375 | 100% | | 7176 | 100% | | 10 551 | 100% |
| Age <45 | 41 | 1.2% | 28.7% | 102 | 1.4% | 71.3% | 143 | 1.4% |
| Age 45–65 | 350 | 10.4% | 26.7% | 963 | 13.4% | 73.3% | 1313 | 12.4% |
| Age 65–75 | 736 | 21.8% | 28.5% | 1848 | 25.8% | 71.5% | 2584 | 24.5% |
| Age >75 | 2248 | 66.6% | 34.5% | 4263 | 59.4% | 65.5% | 6511 | 61.7% |

A) Number of patients per age group.

B) Number of patients per age group in relation to the total number of registered, unregistered or total CKD patients, expressed in percentage (%).

C) Number of registered and unregistered patients per age group in relation to the total number of CKD patients per age group, expressed in percentage (%).

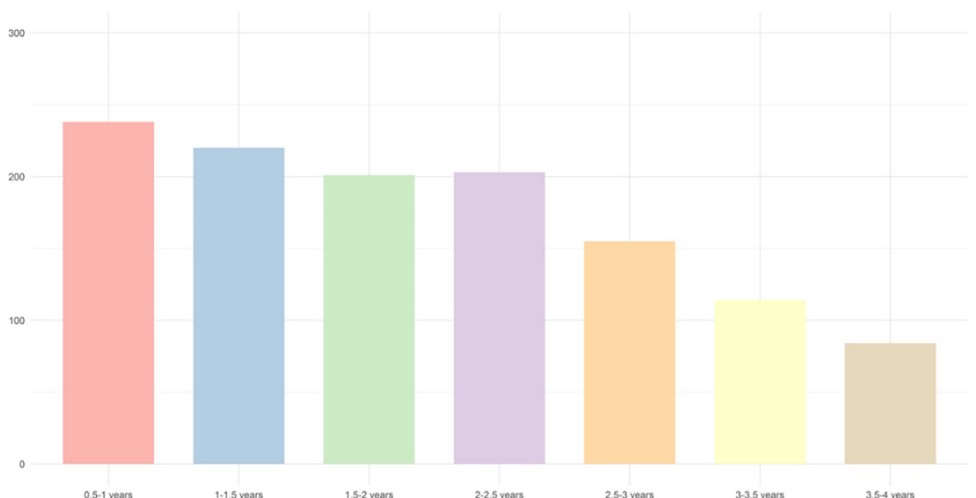

**Fig 3. Stratification of diagnostic delay patients per group.** Y-axis: Number of patients per group. X-axis: diagnostic delay groups.

kidney disease, overactive bladder, renal cyst,. . . the unregistered with a U99 code at baseline did not correspond to chronic insufficiency. However, 30% of the patients with a reduced eGFR and a diagnostic code for CKD also had an other, unspecified kidney disease, while among the unregistered this percentage was much smaller (4.3%). Gout, atrial fibrillation, heart failure, coronary disease and stroke were also common in both registered and unregistered groups.

Results for hemoglobin <8 and 8–10, potassium <5 and >6, ARNI and MRA drugs, alpha blockers, DDP-4 inhibitors, GLP-1RA, SGLT-2 inhibitors, myocardial infarction, hyperkalaemia and covid-19 infection were not significant.

## Multivariate analysis

Being a male, a concurrent diagnosis of diabetes type 1 and 2 (ICPC2 T89 and T90), stroke (ICPC2 K90), heart failure (ICPC2 K77) and hypertension (ICPC2 K86), and more severe CKD (stages 3b, 4 and 5) independently increased the chance on registered CKD (Figs 4 and 5). CKD stage 3a is the reference value. The association was strongest for CKD stage 4 and 5. For myocardial infarction (ICPC2 K75), age and (ex-)smoker, the association was not statistically significant after multiple imputation. Results were shown both before and after multiple imputation (Figs 4 and 5).

## Discussion

In our study, the prevalence of CKD was 5.5%. Among those patients with CKD, a significant amount (68%) did not get a diagnostic code in their EHR. The majority of CKD patients was in stage 3, with a higher proportion registered in stage 5 (75.7% registered) compared to stage 3a (22.9% registered). We found a mean diagnostic delay of 1.94 years (23.3 months), with a maximum follow-up of 3.97 years. Patients with registered CKD took ACEi or ARB and suffered from hypertension and type 2 diabetes more frequently than unregistered at baseline. Being a male, a concurrent diagnosis of diabetes, stroke, heart failure and hypertension, and more severe CKD (stages 3b, 4 and 5) independently increased the chance on registered CKD. The association was strongest for severe kidney failure (CKD stage 4 and 5).

**Table 3. Baseline characteristics.**

| Variable | Registered CKD, n(%) | Unregistered CKD, n(%) | P-value |
|---|---|---|---|
| Total patients | 3375 | 7176 | |
| **Demographics at baseline** | | | |
| Males, n (%) | 1463(43.3%) | 2804(39.1%) | <0.001 |
| Females, n (%) | 1912 (56.7%) | 4372 (60.9%) | <0.001 |
| Mean Age (years) (sd) | 77.6(11) | 76(11.5) | <0.001 |
| 45–65 years, n (%) | 350(10.4%) | 963(13.4%) | <0.001 |
| 65–75 years, n (%) | 736(21.8%) | 1848(25.8%) | <0.001 |
| 75-plus years, n (%) | 2248(66.6%) | 4263(59.4%) | <0.001 |
| Mean Height (cm) (sd) | 166.4(9.7) | 165.6(8.9) | <0.001 |
| Mean Weight (kg) (sd) | 77.1(16.3) | 78(17.3) | <0.001 |
| Mean BMI (kg/m$^2$) (sd) | 28.2(5.2) | 28.6(5.4) | <0.001 |
| Ex-smoker, n (%) | 143 (4.2%) | 207 (2.9%) | <0.001 |
| Smoker, n (%) | 47 (1.4%) | 94 (1.3%) | <0.001 |
| Never-smoker, n (%) | 173 (5.1%) | 299 (4.2%) | <0.001 |
| **Laboratory tests at baseline** | | | |
| Mean hemoglobin (sd) | 12.9(1.6) | 13.1(1.6) | <0.001 |
| hemoglobin < 8 g/100mL, n (%) | 11(0.3%) | 15(0.2%) | 0.259 |
| hemoglobin 8–10 g/100mL, n (%) | 112(3.3%) | 213(3%) | 0.331 |
| hemoglobin 10–12 g/100mL, n (%) | 830(24.6%) | 1360(19%) | <0.001 |
| hemoglobin 12-plus g/100mL, n (%) | 2358(69.9%) | 5470(76.2%) | <0.001 |
| Mean hematocrit (%) (sd) | 39.2(4.6) | 39.9(4.6) | <0.001 |
| Mean egfr (mL/min/1,73 m2) (sd) | 44.4(12.8) | 53.8(10.3) | <0.001 |
| Stage 3A | 1679 (49.7%) | 5699 (79.4%) | <0.001 |
| Stage 3B | 1210 (35.9%) | 1221 (17.0%) | <0.001 |
| Stage 4 | 387 (11.5%) | 210 (2.9%) | <0.001 |
| Stage 5 | 99 (2.9%) | 46 (0.6%) | <0.001 |
| Mean potassium (sd) | 4.7(0.6) | 4.6(0.6) | <0.001 |
| potassium <5.0 mmol/L, n (%) | 2136(63.3%) | 4674(65.1%) | 0.065 |
| potassium 5.0–5.5 mmol/L, n (%) | 602(17.8%) | 892(12.4%) | <0.001 |
| potassium 5.5–6.0 mmol/L, n (%) | 163(4.8%) | 206(2.9%) | <0.001 |
| potassium 6.0–6.5 mmol/L, n (%) | 38(1.1%) | 57(0.8%) | 0.093 |
| potassium 6.5-plus mmol/L, n (%) | 18(0.5%) | 39(0.5%) | 0.947 |
| Mean uACR (mg/g creatinine) (sd) | 248.6(615) | 137.7(390) | <0.001 |
| Mean uPCR (mg/g creatinine) (sd) | 47.6(38.1) | 57.4(29.9) | <0.001 |
| Mean uric_acid (mg/dL) (sd) | 6.8(1.8) | 6.3(1.6) | <0.001 |
| Mean total cholesterol (mg/dL) (sd) | 174.5(41.8) | 177.8(41.9) | <0.001 |
| Mean hdl cholesterol (mg/dL) (sd) | 53.3(16.5) | 54.5(16.3) | <0.001 |
| Mean ldl cholesterol (mg/dL) (sd) | 95(35) | 97.7(35.9) | <0.001 |
| Mean triglyceriden (mg/dL) (sd) | 143.1(80) | 141.6(79.7) | <0.001 |
| **Prescriptions at baseline** | | | |
| ACE inhibitors, n (%) | 1027(30.4%) | 1951(27.2%) | <0.001 |
| ARB drugs, n (%) | 750(22.2%) | 1370(19.1%) | <0.001 |
| ARNI drugs, n (%) | 11(0.3%) | 29(0.4%) | 0.542 |
| MRA drugs, n (%) | 304(9%) | 626(8.7%) | 0.631 |
| Loop diuretics, n (%) | 740(21.9%) | 1177(16.4%) | <0.001 |
| Beta blockers, n (%) | 1631(48.3%) | 3199(44.6%) | <0.001 |
| Thiazide diuretics, n (%) | 0(0%) | 0(0%) | <0.001 |

*(Continued)*

**Table 3.** (Continued)

| Variable | Registered CKD, n(%) | Unregistered CKD, n(%) | P-value |
|---|---|---|---|
| Alpha blockers, n (%) | 4(0.1%) | 6(0.1%) | 0.587 |
| Metformins, n (%) | 506(15%) | 1213(16.9%) | 0.013 |
| Sulfonylurea, n (%) | 238(7.1%) | 434(6%) | 0.049 |
| DDP-4 inhibitors, n (%) | 128(3.8%) | 228(3.2%) | 0.103 |
| GLP-1RA, n (%) | 29(0.9%) | 66(0.9%) | 0.136 |
| Insulins, n (%) | 264(7.8%) | 365(5.1%) | <0.001 |
| Other OAD, n (%) | 56(1.7%) | 64(0.9%) | <0.001 |
| SGLT-2 inhibitors, n (%) | 9(0.3%) | 58(0.8%) | 0.052 |
| Lipid lowering drugs, n (%) | 1653(49%) | 3101(43.2%) | <0.001 |
| Platelet aggregation inhibitors, n (%) | 1126(33.4%) | 2182(30.4%) | 0.002 |
| Anticoagulants, n (%) | 675(20%) | 1289(18%) | 0.012 |
| **Comorbidities at baseline** | | | |
| Myocardial infarction (ICPC2 K75), n (%) | 30(0.9%) | 50(0.7%) | 0.289 |
| Angina pectoris (ICPC2 K74), n (%) | 331(9.8%) | 545(7.6%) | <0.001 |
| Stroke (ICPC2 K90), n (%) | 315(9.3%) | 521(7.3%) | <0.001 |
| Coronary atherosclerosis (ICPC2 K76), n (%) | 326(9.7%) | 476(6.6%) | <0.001 |
| Peripheral vascular disease (ICPC2 K92), n (%) | 309(9.2%) | 459(6.4%) | <0.001 |
| Heart failure (ICPC2 K77), n (%) | 467(13.8%) | 689(9.6%) | <0.001 |
| Atrial fibrillation (ICPC2 K78), n (%) | 775(23%) | 1263(17.6%) | <0.001 |
| Hypertension (ICPC2 K86), n (%) | 2174(64.4%) | 3707(51.7%) | <0.001 |
| Hypertensive kidney failure (ICPC2 K87), n (%) | 147(4.4%) | 129(1.8%) | <0.001 |
| Type 1 Diabetes (ICPC2 T89), n (%) | 109(3.2%) | 128(1.8%) | <0.001 |
| Type 2 Diabetes (ICPC2 T90), n (%) | 1118(33.1%) | 2024(28.2%) | <0.001 |
| Hyperkalemia (ICPC2 T99), n (%) | 104(3.1%) | 189(2.6%) | 0.192 |
| Glomerulonephritis (ICPC2 U88), n (%) | 101(3%) | 98(1.4%) | <0.001 |
| Unspecified kidney disease (ICPC2 U99), n (%) | 1013(30%) | 309(4.3%) | <0.001 |
| Poly cystic Kidney disease (ICPC2 U85), n (%) | 68(2%) | 64(0.9%) | <0.001 |
| Systemic Lupus erythematosus (ICPC2 L99), n (%) | 104(3.1%) | 170(2.4%) | 0.032 |
| Gout (ICPC2 T92), n (%) | 528(15.6%) | 723(10.1%) | <0.001 |
| Covid-19 infection (virus identified) (ICPCC 2 A77), n (%) | 1(0%) | 7(0.1%) | 0.237 |
| Covid-19 infection (virus not identified) (ICPC2 R80), n (%) | 10(0.3%) | 38(0.5%) | 0.097 |

P value was calculated using chi-square for categorical variables and t-test for continuous variables. Urine albumin creatinine ratio (uACR), Urine total protein creatinine ratio (uPCR), Angiotensin-converting enzyme inhibitor (ACE inhibitor), Angiotensin II receptor blocker (ARB), Angiotensin receptor-neprilysin inhibitor (ARNI), Aldosterone receptor agonist (MRA), Glucagon-like peptide-1 receptor agonist (GLP-1 RA), Dipeptidyl peptidase 4 inhibitor (DPP-4 inhibitor), Sodium-Glucose Cotransporter 2 inhibitor (SGLT-2 inhibitors).

Unfortunately, a similar high percentage of unregistered patients as in our study was described by Ryan et al (74%) in 2007 [13]. Our finding about the majority of patients in stage 3 was also consistent with previous reported results [3]. As mentioned before, a large proportion of the patients with early-stage CKD remained without official diagnosis (77.1%). These findings confirm the hypothesis that the diagnosis of CKD is more likely to be made in severe stages of the disease. More worryingly, these results may be an underestimation, as the included practices are among the best registering [19]. Several previous studies have announced that early detection and treatment of CKD is necessary, because it can influence further deterioration of kidney function and it reduces the risk of complications and the number of patients developing ESRD [16, 17]. However, we have to ask ourselves whether not

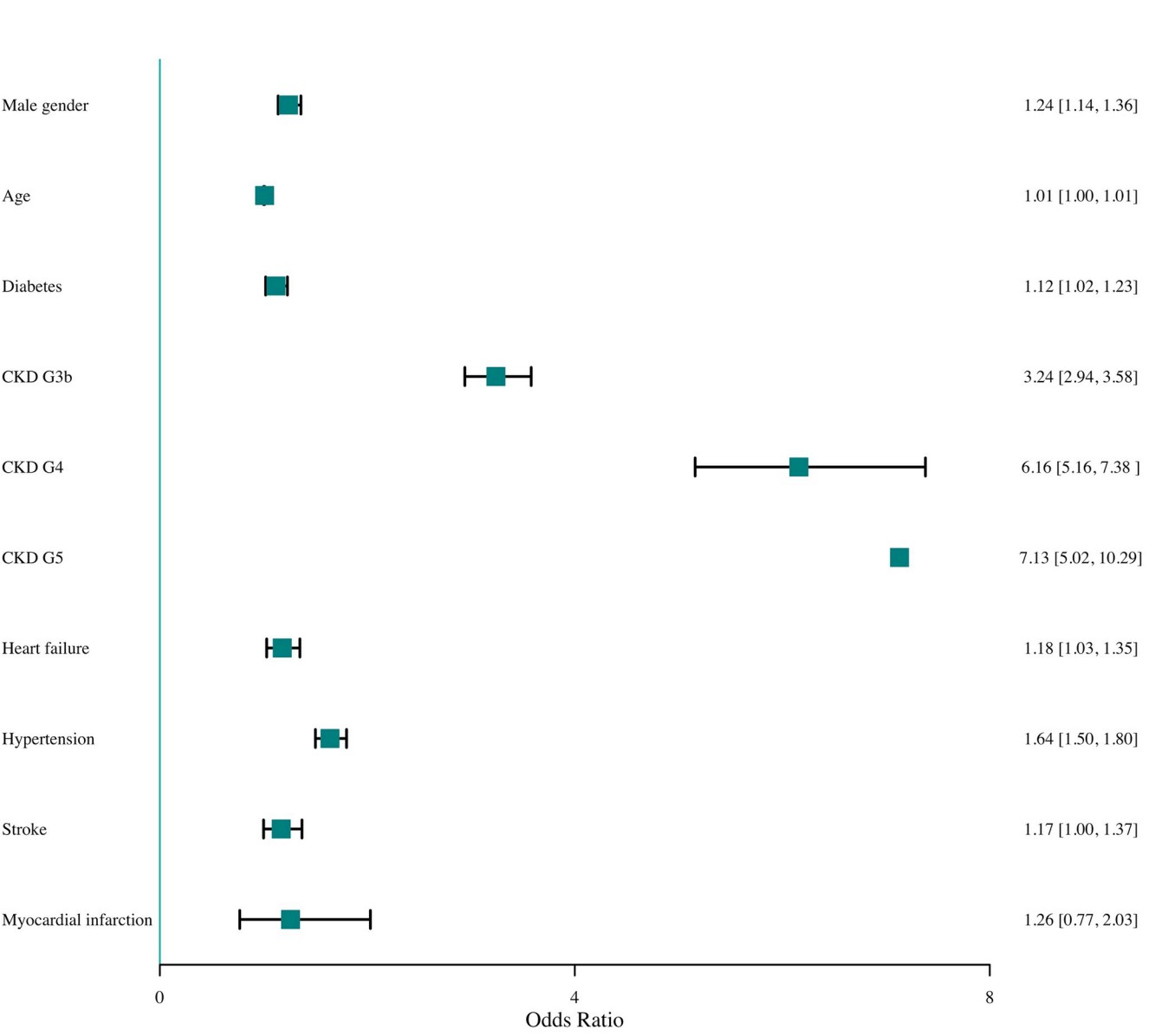

**Fig 4. Odds ratio (OR) and the 95% confidence interval (95% CI) for different variables without smoking (complete analyses).**

registering a CKD diagnosis in the EHR has a similar effect on renal function and complications as not detecting. After all, no diagnostic code does not absolutely imply that the GP is not aware of the kidney problem. It is possible that a proper management and follow-up has been instituted, even though the diagnosis was not registered. However, it is certain that the absence of a diagnostic code in the EHR means that the diagnosis may be missing in referral letters to specialists and lead to the continuity of care being compromised. As a result, renal function may not be considered when the patient presents to the on-call GP or the emergency room. The effect of under-registration on hard endpoints like myocardial infarction should be explored in a subsequent study.

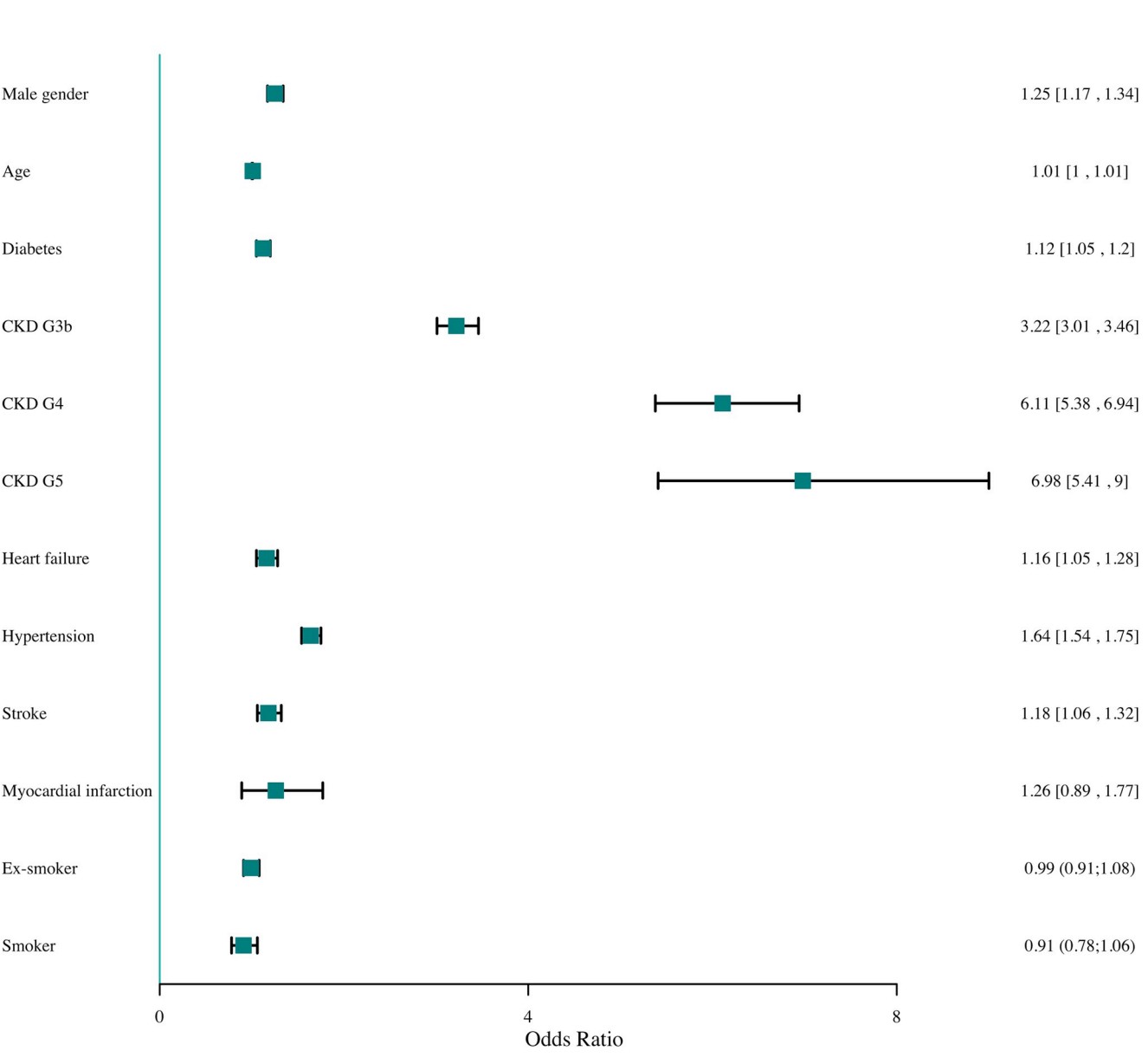

**Fig 5. Odds ratio (OR) and the 95% confidence interval (95% CI) for different variables with smoking (multiple imputation).**

The prevalence of possible CKD in the current study (5.5%) was much lower than earlier published [3, 13]. Previous research in our own INTEGO-database also showed a higher prevalence of CKD (13%) [24]. An explanation for this difference can be found in the study design and selected population. The current study included all patients ≥18 years old, while other studies only used data about patients ≥45 years old [24]. In some patients only one eGFR <60 mL/min/1.73m$^2$ or even no eGFR measurement was available. These patients may also belong to the CKD group, which could result in a higher prevalence. Besides, we did not take the presence of proteinuria into account in the detection of CKD patients. Mainly due to the lack of

data on proteinuria, which brings us straight to the problem of under detection of proteinuria in the Flemish general practice. As a consequence, we underestimated the prevalence among those patients. Both overestimation and underestimation due to single time-point detection of abnormal kidney function and selection of high-risk groups were previously described to explain the difference in prevalence [3, 25, 26].

A recent Iranian study examined the diagnostic delay in CKD patients, but revealed shorter delays (6.5 months) than we reported (23.3 months) [27]. However, the study design and inclusion criteria differed to our study. No other published study on this issue was found. As a result, we are unable to compare our findings with other research. However, a diagnostic delay of 1.94 years may also mean a postponement of correct treatment and follow-up. Further research is necessary to confirm this hypothesis. Remarkable, the mean diagnostic delay was the longest in CKD stage 4 (2.10 years). This means that in patients with severe renal failure valuable time was lost before a diagnostic code was assigned.

Among the patient characteristics, we noticed the mean age for both registered and unregistered was above 75 years. Ryan et al. reported slightly younger mean ages (67.7 years in confirmed CKD and 70.2 years in unconfirmed CKD). As suspected, the mean age was above 65 years in both studies. After all, renal function gradually declines with increasing age in healthy humans [28]. This was also reflected in our results as the largest number of patients were in the group with age >75. As mentioned before in this study, a slightly larger number of patients with registered CKD (52.6%) received an ACEi or ARB compared to the unregistered CKD patients (46.3%). The small difference shows that an ACEi or ARB had been prescribed for unregistered patients, which may suggest that the GP paid attention to the kidney function even though there was no diagnostic code in the EHR. Not all patients from our Intego population received an ACEi or ARB. This finding is not unusual considering the Belgian guidelines on prescribing these drugs in CKD patients [20]. Hypertension (ICPC2 K86) was more frequently present in the registered (64.4%) compared to the unregistered (51.7%), which was not consistent with the results of Ryan et al. (25% in the confirmed and 47% in the unconfirmed) [13]. Further, 33.1% of registered patients had a diagnostic code of type 2 diabetes (T90) at baseline, whereas 28.2% of unregistered patients had. Ryan et al. described much higher percentages in the registered group (59.26%), but similar results in the unregistered (29.33%) [13]. Cardiovascular diseases were common in both registered and unregistered groups. These results suggest that patients with renal insufficiency often experience multimorbidity at baseline, with the presence of differences between registered and unregistered patients. No other studies on predictors of (un)registered CKD could be found.

This research found that a large group of patients never got a diagnosis in their EHR. There seemed to be significant differences between registered and unregistered CKD patients. After all, a registered diagnosis can draw attention while prescribing medication, can improve the efficiency of communication with colleagues, it can facilitate the selection of risk groups. So, thinking about solutions to make it easier to register a diagnosis in the EHR is imperative. The use of a tool that supports registration in the EHR, for example, could offer a possible solution. However, it seems necessary in this regard to find out what reasons GPs state for not registering in order to subsequently make adjustments to the EHR.

Finally, there were some limitations to note. The study used healthcare data to determine the prevalence, which may underrepresent the healthy and asymptomatic that do not seek healthcare. Although the patient population is representative for the Flemish population, registering general practitioners are not representative for the general practitioner population. It is a selected group of high quality registering practitioners which use a specific electronic health record. This selection bias of general practitioners could eventually have an influence on some process parameters in the follow-up of patients [19]. In addition, data collected in a real-world

setting may lack information on specific covariates and laboratory investigations. Data from hospitals and specialist were partly missing. Lab results from the hospital are automatically entered into the EHR, but their diagnoses are not. The number of missing data can be consulted in the S1 Table. We used imputation to fill in the missingness (see method section).

## Conclusion

The proportion of CKD patients stage 3–5 who had no registered CKD code in the EHR was substantial (68%). A registered diagnosis of CKD was more likely to be made in severe stages of the disease (stage 3b, 4 and 5). Being a male, a concurrent diagnosis of diabetes, heart failure and hypertension also independently lowered the risk for unregistered CKD. The differences between registered and unregistered make thinking about solutions to facilitate registration in the EHR imperative.

## Supporting information

**S1 Table. Missing data.**
(DOCX)

## Author Contributions

**Conceptualization:** Ine Van den Wyngaert, Pavlos Mamouris, Gijs Van Pottelbergh.

**Data curation:** Pavlos Mamouris.

**Formal analysis:** Ine Van den Wyngaert, Pavlos Mamouris.

**Funding acquisition:** Bert Vaes, Gijs Van Pottelbergh.

**Investigation:** Pavlos Mamouris.

**Methodology:** Ine Van den Wyngaert, Pavlos Mamouris, Gijs Van Pottelbergh.

**Project administration:** Bert Vaes, Gijs Van Pottelbergh.

**Resources:** Gijs Van Pottelbergh.

**Supervision:** Bert Vaes, Gijs Van Pottelbergh.

**Visualization:** Ine Van den Wyngaert.

**Writing – original draft:** Ine Van den Wyngaert.

**Writing – review & editing:** Ine Van den Wyngaert, Pavlos Mamouris, Bert Vaes, Gijs Van Pottelbergh.

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
