## [Decision Letter · Decision Letter 0]

24 Aug 2022

PONE-D-22-17930An exploration of under-registration of chronic kidney disease in Belgian general practices using logistic regression.PLOS ONE

Dear Dr. Van den Wyngaert,

Thank you for submitting your manuscript to PLOS ONE. After careful consideration, we feel that it has merit but does not fully meet PLOS ONE’s publication criteria as it currently stands. Therefore, we invite you to submit a revised version of the manuscript that addresses the points raised during the review process.

Please see the comments of one reviewer below. Please note that we have only been able to secure a single reviewer to assess your manuscript. We are issuing a decision on your manuscript at this point to prevent further delays in the evaluation of your manuscript. Please be aware that the editor who handles your revised manuscript might find it necessary to invite additional reviewers to assess this work once the revised manuscript is submitted. However, we will aim to proceed on the basis of this single review if possible. 

We look forward to receiving your revised manuscript.

Kind regards,

Hanna Landenmark

Staff Editor

PLOS ONE

Journal Requirements:

a) Did participants provide their written or verbal informed consent to participate in this study?

"No authors have competing interests."

6. Please upload a copy of Supporting Information Table 1 which you refer to in your text on page 18.

Reviewers' comments:

Reviewer's Responses to Questions

**Comments to the Author**

1. Is the manuscript technically sound, and do the data support the conclusions?

Reviewer #1: Partly

2. Has the statistical analysis been performed appropriately and rigorously? 

Reviewer #1: No

3. Have the authors made all data underlying the findings in their manuscript fully available?

Reviewer #1: Yes

4. Is the manuscript presented in an intelligible fashion and written in standard English?

Reviewer #1: Yes

5. Review Comments to the Author

Reviewer #1: Van den Wyngaert et al analyzed data from the INTEGO database since 2018.

They found that the proportion of unregistered chronic kidney disease (CKD) patients was substantial. They also found that advanced stages of CKD, male, diabetes, heart failure, hypertension were associated with low risks for unregistered CKD. I have some comments about this manuscript.

1. Page 6, lines 108, Methods: In order to correctly define CKD, the calculation of eGFR should be standardized. The authors did not mention the equation that calculated eGFR. According to the KDIGO (as the reference No. 2 of this manuscript) or the KDOQI guidelines, eGFR should be calculated using the MDRD or 2009 CKD-EPI equation. A newer 2021 CKD-EPI equation has also been suggested recently.

2. Page 6, lines 108, Methods: the authors only included stages 3-5 CKD because they only include those who had an eGFR < 60 ml/min/1.73m2. This information should be more clearly described throughout the manuscript from the title, abstract, and manuscript text. Otherwise, it may cause misinterpretation that the study is providing information including the whole range of CKD patients.

3. Page 6, lines 111-112, Methods: The author used data since 2018, but did not mention the date for end of follow-up of the data.

4. Page 7, lines 127-129, Methods: This section provides almost the same information as line 107-111.

5. Page 8, lines 145-146, Methods: This description does not seem to be the calculation of "median time", should be explained more clearly. In addition, this sentence seems to provide very similar information as line 147-149.

6. Page 8, lines 160-161, Methods: The author did not mention the process of model selection for the multivariate logistic regression, and how to decide the final model as they presented in the Results section. Since the INTEGO database provided abundant information and covariates, how did author decide the inclusion and exclusion of these covariates? In addition, some of the covariates included in the multivariate model were not provided in the baseline descriptive analyses (Table 3) and some of the covariates did not have a clear definition (ex. did the covariate of diabetes include both type 1 & 2 diabetes? Did the covariate of hypertension included hypertensive kidney failure?)

7. Pages 8-9, lines 162-171, Methods: The imputation method for missing variables was not clearly described. The imputation procedure seemed to be based on unpublished data (Reference No. 22), and the detailed imputation method should be clearly and comprehensively described in the current manuscript unless this imputation method has been published. How many covariates has been imputed? What is the percentage of missing data in the study cohort? A comparison before and after imputation should be provided. The Supporting information (S1 Table. Missing data) was not provided with the submitted manuscript. In addition, what is an extra covariate?

8. Page 11, lines 223-226, Results: There seemed to be difference between registered and unregistered groups for diuretics, lipid lowering drugs, platelet aggregation inhibitors, and anticoagulants.

9. Page 11, lines 228-230, Results: the description regarding diagnosis code U99 provide duplicate information as in the Methods section (lines 117-120).

10. Table 3: There were 309 (4.3%) of unregistered CKD patients who had a diagnosis code of "Unspecified kidney disease (ICPC2 U99)" at baseline which is confusing. Please explain.

6. PLOS authors have the option to publish the peer review history of their article (what does this mean?). If published, this will include your full peer review and any attached files.

Reviewer #1: No

---

## [Author Response · Author response to Decision Letter 0]

27 Oct 2022

Dear Editor, 

Dear reviewers,

Please find enclosed a revised version of our manuscript entitled “An exploration of under-registration of chronic kidney disease in Belgian general practices using logistic regression”.

We gratefully thank you for considering our manuscript for publication and providing these important comments. We have studied your comments and suggestions carefully and we have adapted the manuscript accordingly. We truly believe that this process has improved the quality of the manuscript substantially.

We made some adjustments based on the editor’s comments:

File naming was adjusted according to PLOS ONE’s requirements.

a) Did participants provide their written or verbal informed consent to participate in this study?

The INTEGO procedure was approved by a written consent by the ethical review board of the Medical School of KU Leuven (N° ML 1723) and by the Belgian Privacy Commission (no SCSZG/13/079). Written consent was obtained from every of the participating Intego primary care practices. 

The code was attached in a separate word file. 

"No authors have competing interests."

We have no Competing Interests. The following statement was added in the cover letter: "The authors have declared that no competing interests exist."

As stated when submitting, it is not possible to make a minimum dataset publicly available, but data requests can be made to the Data Protection Officer (Sciensano) . 

The data set necessary to validate our analysis contains identifying patient-level data which cannot be suitably de-identified or aggregated and can therefore only be accessed inside a monitored analysis environment. These restrictions were imposed by the Belgian National Information Security Committee: section Social Security and Health: 

°Websites (only FR and NL):https://ehealth.fgov.be/ehealthplatform/nl/informatieveiligheidscomite
https://ehealth.fgov.be/ehealthplatform/fr/comite-de-securite-de-linformation oMain deliberations (only FR and NL): 

- Beraadslaging nr. 15/009 van 17 februari 2015 betreffende de generieke methode voor de uitwisseling van gecodeerde persoonsgegevens die de gezondheid betreffen in het kader van healthdata.be en healthstat.be (https://ehealth.fgov.be/ehealthplatform/file/view/23acd078e8a4cb025e216b6d24906ae 480b57cf8?filename=15-009-n023-healthdata_be.pdf) 

- Délibération n° 15/009 du 17 février 2015 relative à la méthode générique d'échange de données à caractère personnel codées relatives à la santé, dans le cadre de healthdata.be et healthstat.be. (https://ehealth.fgov.be/ehealthplatform/file/view/ea116a489cc473f8174c8559dd060b4f aa3ff651?filename=15-009-f023-healthdata_be.pdf) 

° INTEGO deliberations (only FR and NL): 

- Beraadslaging nr. 13/026 van 19 maart 2013, laatst gewijzigd op 7 april 2020, met betrekking tot de mededeling van gepseudonimiseerde persoonsgegevens die de gezondheid betreffen, via het healthdata.be platform, in het kader van de samenstelling, het gebruik en de ter beschikking stelling van een register voor epidemiologisch onderzoek (Intego). è https://ehealth.fgov.be/ehealthplatform/file/view/AXGB3PooVF2eEzQeBL5P?filename= 13-026-n162-INTEGO%20Covid19-gewijzigd%20op%207%20april%202020.pdf

- Délibération n° 13/026 du 19 mars 2013, modifiée en dernier lieu le 7 avril 2020, relative à la communication de données à caractère personnel pseudonymisées relatives à la santé, via la plateforme healthdata.be, dans le cadre de la constitution, de l'utilisation et de la mise à la disposition d'un registre de recherches en épidémiologie (Intego). (https://ehealth.fgov.be/ehealthplatform/file/view/AXGB2dtUmTlaOSp4Nmk3?filename= 13-026-f162-INTEGO%20Covid19-modifi%C3%A9e%20le%207%20avril%202020.pdf)

Data requests may be sent to Mr. Roel Heijlen, Data Protection Officer healthdata.be (Sciensano), Roel.Heijlen@sciensano.be, Rue Juliette Wytsmanstraat 14, 1050 Brussels. Interested researchers will need to provide their name, first name, professional organization name, email address, mobile number and the database of interest to request access.

6. Please upload a copy of Supporting Information Table 1 which you refer to in your text on page 18.

Supporting Information Table 1 was added. 

We also made some adjustments based on the reviewer’s comments:

1. Page 6, lines 108, Methods: In order to correctly define CKD, the calculation of eGFR should be standardized. The authors did not mention the equation that calculated eGFR. According to the KDIGO (as the reference No. 2 of this manuscript) or the KDOQI guidelines, eGFR should be calculated using the MDRD or 2009 CKD-EPI equation. A newer 2021 CKD-EPI equation has also been suggested recently.

Thank you for this comment. Not all eGFR were calculated by the same equation due to the design of the database, which collects data from practices throughout Flanders. However, we selected these either based on MDRD or CKD-EPI. We added this in the manuscript.

2. Page 6, lines 108, Methods: the authors only included stages 3-5 CKD because they only include those who had an eGFR < 60 ml/min/1.73m2. This information should be more clearly described throughout the manuscript from the title, abstract, and manuscript text. Otherwise, it may cause misinterpretation that the study is providing information including the whole range of CKD patients.

Thank you for this suggestion. This was added in the title, abstract, introduction and conclusion.

3. Page 6, lines 111-112, Methods: The author used data since 2018, but did not mention the date for end of follow-up of the data.

We should have mentioned the date for end of follow-up of the data. This was 31/03/2021. It was added in the manuscript.

4. Page 7, lines 127-129, Methods: This section provides almost the same information as line 107-111.

Thank you for this comment. The inclusion criteria section was modified with reference to the study population section. 

5. Page 8, lines 145-146, Methods: This description does not seem to be the calculation of "median time", should be explained more clearly. In addition, this sentence seems to provide very similar information as line 147-149.

• Lines 145-146: This was indeed a description of the diagnostic delay and not the median time. This has been adjusted and was described more clearly. 

• Lines 147-149: Similar information was deleted. 

6. Page 8, lines 160-161, Methods: The author did not mention the process of model selection for the multivariate logistic regression, and how to decide the final model as they presented in the Results section. Since the INTEGO database provided abundant information and covariates, how did author decide the inclusion and exclusion of these covariates? In addition, some of the covariates included in the multivariate model were not provided in the baseline descriptive analyses (Table 3) and some of the covariates did not have a clear definition (ex. did the covariate of diabetes include both type 1 & 2 diabetes? Did the covariate of hypertension included hypertensive kidney failure?)

Thank you for this fair comments. We used covariates based on clinical expertise after consultation with experts in the field.

Smoking and eGFR changes were left out in the submitted version. We now added them in table 3.

Diabetes included both type 1 & 2, hypertension did not include hypertensive kidney failure. Lines 248-251: We added the ICPC2 codes to clarify the definition of the covariates. 

7. Pages 8-9, lines 162-171, Methods: The imputation method for missing variables was not clearly described. The imputation procedure seemed to be based on unpublished data (Reference No. 22), and the detailed imputation method should be clearly and comprehensively described in the current manuscript unless this imputation method has been published. How many covariates has been imputed? What is the percentage of missing data in the study cohort? A comparison before and after imputation should be provided. The Supporting information (S1 Table. Missing data) was not provided with the submitted manuscript. In addition, what is an extra covariate?

The imputation method was indeed based on unpublished data. Only smoking was missing. Therefore, we imputed the smoking variable for the year 2018, using the method of Multiple Imputation as described by Rubin (1) and implemented in Rpackage mice by Van Buuren (2) using several auxiliary variables including age, gender, socioeconomic status, and relevant diagnoses and prescriptions. The percentage of missing data was added and a figure with results before and after multiple imputation was provided (figure 4 and 5). The supporting information was attached. 

(1) Rubin DB. Multiple Imputation for Nonresponse in Surveys. New York: John Wiley and Sons; 2004.

(2) Van Buuren, S., & Groothuis-Oudshoorn, K. (2011). mice: Multivariate imputation by chained equations in R. Journal of statistical software, 45, 1-67.

8. Page 11, lines 223-226, Results: There seemed to be difference between registered and unregistered groups for diuretics, lipid lowering drugs, platelet aggregation inhibitors, and anticoagulants.

This is a fair comment. There are indeed differences in percentage between registered and unregistered groups. The word choice was incorrect, so we adjusted it. 

9. Page 11, lines 228-230, Results: the description regarding diagnosis code U99 provide duplicate information as in the Methods section (lines 117-120).

Thank you for this comment. We removed it in the method section, eliminating duplicate information. 

10. Table 3: There were 309 (4.3%) of unregistered CKD patients who had a diagnosis code of "Unspecified kidney disease (ICPC2 U99)" at baseline which is confusing. Please explain.

As explained at line 231-235, patients with unspecified kidney disease are known to have a urinary tract disease other than renal insufficiency like overactive bladder, renal cyst,… Unfortunately, the ICPC2 code for chronic kidney disease is the same as the code for unspecified kidney disease, namely U99. Some of the patients with unspecified kidney disease also had decreased renal function, so they should also have a diagnostic code of CKD.

Sincerely,

Ine Van den Wyngaert & co-authors

---

## [Decision Letter · Decision Letter 1]

5 Dec 2022

An exploration of under-registration of chronic kidney disease stages 3-5 in Belgian general practices using logistic regression.

PONE-D-22-17930R1

Dear Dr. Van den Wyngaert,

We’re pleased to inform you that your manuscript has been judged scientifically suitable for publication and will be formally accepted for publication once it meets all outstanding technical requirements.

Kind regards,

Tatsuo Shimosawa, M.D., Ph.D.

Academic Editor

PLOS ONE

Additional Editor Comments (optional):

Reviewers' comments:

Reviewer's Responses to Questions

**Comments to the Author**

1. If the authors have adequately addressed your comments raised in a previous round of review and you feel that this manuscript is now acceptable for publication, you may indicate that here to bypass the “Comments to the Author” section, enter your conflict of interest statement in the “Confidential to Editor” section, and submit your "Accept" recommendation.

Reviewer #2: All comments have been addressed

Reviewer #3: All comments have been addressed

2. Is the manuscript technically sound, and do the data support the conclusions?

Reviewer #2: Yes

Reviewer #3: Yes

3. Has the statistical analysis been performed appropriately and rigorously? 

Reviewer #2: Yes

Reviewer #3: Yes

4. Have the authors made all data underlying the findings in their manuscript fully available?

Reviewer #2: Yes

Reviewer #3: Yes

5. Is the manuscript presented in an intelligible fashion and written in standard English?

Reviewer #2: Yes

Reviewer #3: Yes

6. Review Comments to the Author

Reviewer #2: Early recognition of CKD and appropriate medical intervention are important in terms of reducing the number of patients with end-stage renal failure and preventing cardiovascular disease. This manuscript is valuable because it reveals a significant number of patients with CKD 3-5 whose CKD codes are not registered in the HER, indicating the importance of intervention in this patient group.

In the future, it is hoped that measures will be taken to facilitate enrollment in the HER and to provide appropriate interventions for CKD patients who are relatively unnoticed due to their lack of comorbidities.

Reviewer #3: In the present observational study, the Authors described prevalence of unregistered CKD in a Belgian General Practitioner population. Among included patients, there were 10 551 patients (5.5%) with two consecutive eGFR laboratory measurements indicating chronic kidney disease (CKD) (eGFR <60 mL/min/1.73m2), recorded at least three months apart during the baseline period. Among those patients with CKD, a significant amount (68%) did not get a diagnostic code in their Electronic Health Record (HER). Being a male, a concurrent diagnosis of diabetes, heart failure and hypertension also independently increased the possibility of a correctly registered CKD.

The Authors conclude that thinking about solutions to make it easier to register a diagnosis in the EHR is imperative.

The revised version of the paper is fairly well written, the aim is an interesting one, the methodological approach is appropriate, the results are adequately commented upon and limitations have been correctly examined.

7. PLOS authors have the option to publish the peer review history of their article (what does this mean?). If published, this will include your full peer review and any attached files.

Reviewer #2: No

Reviewer #3: No

---

## [Editor Report · Acceptance letter]

15 Dec 2022

PONE-D-22-17930R1 

An exploration of under-registration of chronic kidney disease stages 3-5 in Belgian general practices using logistic regression 

Dear Dr. Van den Wyngaert:

I'm pleased to inform you that your manuscript has been deemed suitable for publication in PLOS ONE. Congratulations! Your manuscript is now with our production department. 

Kind regards, 

on behalf of

Prof. Tatsuo Shimosawa 

Academic Editor

PLOS ONE